# Predictors of Survival in Under-Five Children with Low Birth Weight: A Population-Based Study in Indonesia

**DOI:** 10.3390/nursrep15070238

**Published:** 2025-06-29

**Authors:** Eka Mishbahatul Marah Has, Ferry Efendi, Sylvia Dwi Wahyuni, Novianti Lailiah, Rio Arya Putra Mahendra

**Affiliations:** 1Department of Advanced Nursing, Faculty of Nursing, Universitas Airlangga, Surabaya 60115, Indonesia; ferry-e@fkp.unair.ac.id; 2School of Nursing and Midwifery, La Trobe University, Melbourne 3086, Australia; 3Research Center in Advancing Community Healthcare, Surabaya 60115, Indonesia; 4Department of Basic Nursing, Faculty of Nursing, Universitas Airlangga, Surabaya 60115, Indonesia; sylvia.dwiwahyuni@fkp.unair.ac.id; 5Nursing Study Program, Faculty of Medicine, Universitas Negeri Surabaya, Surabaya 60213, Indonesia; noviantilailiah@unesa.ac.id; 6Nursing Study Program, Faculty of Nursing, Universitas Airlangga, Surabaya 60115, Indonesia; rioaryaputram@gmail.com

**Keywords:** survival analysis, child mortality, child health, low birth weight, Indonesian DHS

## Abstract

**Background/Objectives:** The under-five mortality rate (U5MR) remains a serious health challenge in Indonesia, with low birth weight (LBW) being a key risk factor. This study aimed to identify predictors of survival among under-five children with LBW using data from the 2017 Indonesia Demographic and Health Survey (IDHS). **Methods:** This cross-sectional study included 625 children under five with LBW. The dependent variable was under-five mortality (children aged 0–59 months), while the independent variables include child (gender), maternal (age at delivery, education, empowerment, delivery complications, and breastfeeding history), health service (antenatal care-ANC and place of delivery), and household (wealth quintile and residence) factors. Data were obtained from the 2017 IDHS using household and women’s questionnaires and analyzed using univariate analysis, Kaplan–Meier estimation, and Cox regression. **Results:** 41 of 625 children born with LBW died before the age of five. The Kaplan–Meier estimation found that maternal (delivery complications and breastfeeding history), health service (ANC history and place of delivery), and household factors (residence) significantly influenced the survival of LBW children. The Cox regression results showed that LBW children who were breastfeed and whose mother had adequate antenatal care visits had a lower risk of under-five mortality. Surprisingly, children born in health facilities had a significantly higher risk of death compared to those born elsewhere. **Conclusions:** Exclusive breastfeeding, adequate antenatal care, and place of delivery are important determinants of survival among children born with LBW. This support targeted interventions to improve the survival chances of children born with LBW, particularly in their early years of life.

## 1. Introduction

The under-five mortality rate (U5MR) is a key indicator of child health and well-being. Despite global progress, the reduction in the U5MR worldwide has recently slowed [1]. In 2023, the global U5MR decreased by 61 percent, from 94 deaths per 1000 live births in 1990 to 37. Approximately 4.8 million children under five died this year, translating to around 13,100 deaths per day [2]. Nearly half (47%) of these deaths occurred during the neonatal period-defined as the first 28 days of life-highlighting the increasing burden of neonatal mortality, which contributes disproportionately to under-five death [3,4]. Similarly, Indonesia’s U5MR declined significantly from 83.9 in 1990 to an estimated 20.6 in 2023 [5]. However, it remains a country in Southeast Asia with the 7th highest neonatal mortality rate (10.53 deaths per 1000 live births) by 2023 [6]. Alarmingly, data from Indonesian Health Survey 2023 recorded a sharp increase in neonatal death, from 20,882 in 2022 to 29,945 in 2023 [7]. If this trend persists, Indonesia is at risk of missing the Sustainable Development Goals’ (SDGs) targets for both the U5MR and neonatal mortality by 2030. These figures emphasize the urgent need for targeted interventions that enhance the survival of neonates, a critical factor in reducing under-five mortality.

Being born with low birth weight (LBW) significantly increases the risk of under-five mortality [8,9,10]. According to the World Health Organization (WHO), LBW is defined as a weight at birth less than 2500 g, regardless of gestational age [11]. It is considered one of the strongest predictors of child survival [12]. Children born with LBW are almost 30 times more likely to die within the first year of life [13], due to their underdeveloped organ and weak immune system, which make them highly vulnerable to infections. Additionally, congenital defects and related complications can further compromise their health and development, increasing the long-term burden on child health outcomes [14]. The Indonesia Demographic and Health Survey (IDHS) 2017 reported that the national average percentage of children born with LBW was 6.98% [15]. This figure increased to 11.7% in 2021, with a higher prevalence observed in rural areas [7,16].

Proper care is essential for children born with LBW to enhance their chance of survival [17,18]. However, many of them in Indonesia struggle to reach their 5th birthday, due to a combination of biological vulnerabilities and inadequate health care system [19]. Identifying the key determinants that influence survival among under-five children born with LBW is, therefore, critical to guide the development of effective, targeted intervention.

Mosley and Chen provide a conceptual framework that offers insights into the determinants of child survival. It includes individual-, household-, and community-level factors [20,21]. Previous studies conducted abroad have revealed several maternal factors, such as age at delivery, educational level, birth interval, parity, health insurance coverage, gestational age, type of delivery, exclusive breastfeeding, and immunization, significantly contribute to under-five children’s survival. Other factors, such as place of delivery and place of residence, also influence survival outcomes [22,23,24,25,26]. A “male disadvantage” in survival has been observed, indicating that male children may have poorer survival outcomes compared with females [27]. However, there is a lack of evidence on how these factors specifically affect survival among children with LBW in Indonesia.

While these determinants have been widely studied in other contexts, research in Indonesia remains limited. Despite the significant prevalence of under-five mortality in children born with LBW in the country, few studies have comprehensively examined their survival [12,28,29]. Existing research focusing on LBW populations tends to explore only limited determinants, primarily socioeconomic factors [30], while neglecting key child-, maternal-, household-, and healthcare-related variables. Moreover, there is a lack of analyses utilizing extensive, nationally representative data, such as the 2017 IDHS [31], which is the latest available IDHS dataset to date. This highlights a critical research gap in identifying the broader predictors of survival among under-five children with LBW in the Indonesian context. This study aimed to identify predictors of survival among under-five children with LBW using data from the 2017 IDHS.

## 2. Materials and Methods

### 2.1. Study Design

This study employed a cross-sectional design using secondary data from the 2017 IDHS. The IDHS is cross-sectional and contains retrospective birth history and child mortality data, including information on age at death. It allows for time-to-event analysis, such as survival analysis, with some limitations.

### 2.2. Data Source

The 2017 IDHS is a population-based survey administered by the National Population and Family Planning Board (BKKBN), Statistics Indonesia (BPS), Indonesian Ministry of Health, and ICF International, funded by the United States Agency for International Development (USAID). It provides estimates of key demographic and health indicators, including child mortality. The survey collected information from a representative sample of women aged 15–49 years across all 34 provinces of Indonesia. According to the 2017 census, there were approximately 70.5 million women in this age group, representing 53.6% of the total female population. A two-stage stratified sampling method was applied to ensure national representativeness. Out of 50,730 eligible women identified, 49,627 completed the interviews, resulting in a response rate of 98% [31].

Permission to use the data was obtained from ICF International through an application submitted via the DHS Program’s website (URL: https://www.dhsprogram.com/data/available-datasets.cfm, accessed on 22 May 2025). The dataset was downloaded following approval.

### 2.3. Study Population and Sample

The population of this study consisted of all last live-born children born between July 2012 and September 2017, as recorded in the 2017 IDHS. Children were identified from the birth record (BR) file, which includes birth histories and survival status. The relevant variables used to define age- and birth-related information included date of birth, age at death, survival status, and birth weight. A subset of children under five years of age who met the following inclusion criteria was extracted as the study sample: (1) live-born children; (2) singleton births; and (3) availability of complete data on birth weight, birth status, and age at death. Only those classified as having low birth weight (LBW), defined as a birth weight less than 2500 g regardless of gestational age, from either a written record or mother’s report [11,31], were included in the final analysis.

A total of 707 children met these criteria. After excluding 82 respondents with missing data on key variables, like birth weight and age of death, through listwise deletion, the final sample size was 625 children.

Sampling weights were not applied, as the analysis focused on a specific subpopulation (children born with low birth weight) to explore survival patterns rather than provide nationally representative estimates.

### 2.4. Ethics Considerations

The 2017 IDHS ethics were granted under the Institutional Review Board (IRB) Findings Form ICF IRB FWA00000845. Written consent is obtained from all participants prior to data collection. All personal identifiers were removed from the dataset to ensure confidentiality [31]. More information about the ethical review process is available on the DHS program’s website (URL: https://dhsprogram.com/Methodology/Protecting-the-Privacy-of-DHS-Survey-Respondents.cfm, accessed 22 May 2025).

### 2.5. Variables

The dependent variable in this study was under-five child survival, measured using child mortality data. Under-five mortality refers to the likelihood of a child dying between birth and their fifth birthday (ages 0–4 years) [2]. Data were obtained from the Women’s Questionnaire (17IDHS-W), specifically Section 6 (child health and nutrition).

The independent variables in this study consisted of child, maternal, health service, and household factors. Child factors included gender (female and male). Maternal factors included the following: (1) age at delivery (<35 and ≥35) [32]; (2) educational level (elementary school, secondary/high school, and diploma/university) [33]; (3) empowerment, which is assessed based on the number of household decisions the mother participated in, specifically related to her own’s health care, major household purchases, and visits to her family or relatives (low, medium, and high) [31]; (4) delivery complications, which are assessed from the presence or absence of danger signs of delivery in the last pregnancy (yes and no) [34]; and (5) breastfeeding history, which is assessed on whether the mother has ever breastfed her child (yes and no). Health service factors included the following: (1) antenatal care (ANC) history, which is assessed from the frequency of mothers accessing ANC services in the last pregnancy (<4 and ≥4 times) [35]; and (2) place of delivery (health facilities and elsewhere). Household factors included the following: (1) wealth index (poor, middle, and rich) [31] and (2) residence, which is classified according to Statistics Indonesia statute (urban and rural) [36]. The instrument used is the Households Questionnaire (17IDHS-HH) and the Women’s Questionnaire (17IDHS-W), specifically Section 1 (respondent’s background), Section 2 (birth history), and Section 4 (pregnancy and postnatal care) [31].

### 2.6. Statistical Analysis

A descriptive (univariate) analysis was conducted to summarize the distribution of key variables. Life table analysis was then utilized to estimate the cumulative probability of survival based on birth weight and other selected factors for children under five years of age [37]. The Kaplan–Meier survival analysis was performed to compare survival times between groups, with the mean survival times reported. The log-rank test was applied to assess the statistical significance of differences in survival distributions between groups [38].

Finally, multivariate analysis was performed using Cox proportional hazards regression to identify the factors associated with under-five mortality [39]. The proportional hazards assumption was assessed with log-minus-log survival plots, revealing no substantial violations. Variables with *p* < 0.25 in the bivariate analysis were included in the multivariable model, which was guided by theoretical frameworks and prior empirical findings. The model’s fit was assessed using the Omnibus test. However, specific measures of predictive accuracy, such as Harrell’s C-index or AIC, were not available in SPSS and, therefore, not reported. The strength of association was reported as hazard ratios (HRs) with 95% confidence intervals (CIs).

All statistical analyses were conducted using IBM SPSS statistics version 30 (IBM Corp., Armonk, NY, USA). The significance level for this study was set at 0.05. However, because this study is exploratory, a significance level of 0.10 was also acceptable for finding potentially meaningful connections in future research.

## 3. Results

A total of 625 under-five children born with LBW, drawn from various provinces in Indonesia and recorded in the IDHS, were included in the analysis. Of these, 41 children (6.6%) died before reaching the age of five.

### 3.1. Bivariate Survival Analysis

Table 1 presents the bivariate analysis of under-five survival among children with LBW according to selected child-, maternal-, health-service-, and household-related factors using the Kaplan–Meier survival analysis. Overall the survival percentages and mean survival time (in months) were estimated for each subgroup. The log-rank test was employed to assess differences in survival distributions across groups.

Survival rates for children under five years old with LBW were significantly linked to various maternal, health service, and household factors. Key factors that were statistically significant at the 5% level included breastfeeding history, delivery complications, and the area of residence. Additionally, maternal age at delivery and the place of delivery displayed marginal significance (*p* < 0.10). Please refer to Table 1 for further details.

Children born to mothers who were under 35 years old at the time of delivery experienced a higher survival rate (94.5%) and a longer mean survival time (55.99 months) compared to those whose mothers were 35 years or older (91.1%, 54.17 months; *p* = 0.093). Children delivered without complications had a higher survival rate (96.0%) and longer average survival time (56.95 months) compared to those born with complications (92.0%, 54.52 months; *p* = 0.044).

Breastfeeding history showed the strongest association with children’s survival. Children who ever breastfed had markedly higher survival (97.5%) and longer mean survival time (57.60 months) compared with non-breastfed children (61.4%, 41.90 months; *p* < 0.001). Children whose mothers had four or more ANC visits during pregnancy had a higher survival rate (94.0%) and longer mean survival time (55.68 months) compared with those with fewer than four visits (89.9%, 53.39 months; *p* = 0.096).

Children born in health facilities had a slightly lower mortality rate (7.2%) and mean survival time of 54.86 months, while those delivered elsewhere had higher survival (95.5%) and a longer mean survival time (56.88 months; *p* = 0.093). Lastly, children residing in rural areas showed better survival (95.3%) and longer mean survival time (56.38 months) compared with those in urban areas (91.5%, 54.36 months; *p* = 0.049).

Because most children remained alive throughout the follow-up period, the median survival time could not be estimated, as survival probabilities did not fall below 50% for most groups, suggesting relatively good survival outcomes.

### 3.2. Multivariate Survival Analysis Using Cox Regression

Table 2 highlights the Cox proportional hazards regression model results, which examine the association between selected predictors and under-five mortality in children with LBW. The table includes hazard ratios (HRs), 95% confidence intervals (CIs), and *p*-values. Variables with *p*-values below the 0.10 threshold were considered to have statistically potential associations and are described below.

The Cox regression model demonstrated statistical significance (Omnibus test of model coefficients: χ^2^ = 130.35, df = 13, *p* < 0.001), indicating that it explained a significant amount of the differences in under-five mortality.

Table 2 demonstrated that a lack of breastfeeding is strongly associated with an increased risk of mortality among children born with LBW. Children who were never breastfed had a hazard of death that was 14.58 times higher than those who were breastfed (95% CI: 7.22–29.45, *p* < 0.001). Additionally, children whose mothers had fewer than four ANC visits during pregnancy were 2.40 times more likely to die before reaching their fifth birthday compared to those with adequate ANC visits (95% CI: 0.94–6.11, *p* = 0.067). The place of delivery also showed an association with mortality risk. Surprisingly, children born in health facilities exhibited a 2.39 times higher risk of death compared to those born at home or elsewhere (95% CI: 0.88–6.48, *p* = 0.088).

## 4. Discussion

This study identified several key predictors of survival among under-five children with LBW according to Mosley and Chen Framework [21], using data from the 2017 IDHS, based on both Kaplan–Meier survival analysis and Cox proportional hazards regression. This included maternal factors (age at delivery, delivery complications, and breastfeeding history), health service factors (ANC history and place of delivery), and household factor (area of residence).

### 4.1. Interpretation of Key Findings

Children with LBW who are born to mothers under 35 years of age demonstrated better survival outcomes compared with those born to older mothers. As a mother’s age at delivery increases, the risk of children dying before the age of five also increases [29,40,41]. Advanced maternal age women (≥35 years old) are at a higher risk of adverse obstetrical and perinatal outcomes [42]. This increased risk potentially due to changes in the female reproductive system following aging and the higher prevalence of comorbidities. These factors may compromise fetal growth and neonatal adaptation and, therefore, could negatively impact the survival of children with LBW [43]. This finding is consistent with previous studies indicating that children born from advanced maternal age mother is more prone to neonatal complications, including LBW and other health risks, which increase their morbidity and mortality [12,24,44,45]. As the trend of childbirth at advanced maternal age continues to rise in developing countries [46], including Indonesia, improving the survival of children with LBW born from this mother requires a comprehensive strategy. It includes strengthening preconception and ANC through risk screening, management of chronic conditions, and individualized pregnancy monitoring. Ensuring delivery at well-equipped health facilities, followed by high-quality neonatal care, is also essential to enhance outcomes for this vulnerable group.

Children born with LBW who are delivered without complications have a better chance of survival compared to those who experience complications during delivery. This finding aligns with existing literature, which indicates that delivery complications, such as prolonged labor, fetal distress, or obstructed labor can increase the risk of hypoxia, trauma, and early neonatal morbidity [47]. LBW children also tend to have a low immune system, making them vulnerable to infections. Complications such as premature rupture of membranes (PROM) can increase the likelihood of neonatal sepsis [48,49]. All of which may further compromise neonatal health, delay recovery, and significantly elevate the risk of early mortality [3,50,51]. Although this variable was not significant in multivariate model, the observed bivariate association suggests that delivery-related complications potentially contribute to early life vulnerability. WHO had published comprehensive recommendations for managing complication during pregnancy and childbirth to ensure a positive birth experience [34,52]. Nevertheless, Indonesia continues to face challenge in providing equitable and high-quality obstetric services [53,54]. Strengthening skilled birth attendance, enhancing intrapartum monitoring, and improving referral systems is crucial to improve survival among LBW children in these high-risk conditions.

Breastfeeding emerged as the strongest predictor of survival among children with LBW. Children born with LBW who never breastfed had a significantly higher risk of death compared to those who breastfed at least once. Previous studies highlight similar findings [40,55,56,57]. This supports global evidence that breastfeeding is an essential source of nutrition that promotes healthy physiological and cognitive development while offering immunological benefits to help protect children with LBW from disease challenges and increase their chance of survival [58,59,60]. Human milk is superior in composition as it is rich in carbohydrates, proteins, fats, and several biologically active compounds, such as immunoglobulins, lactoferrin, and lymphocytes, which are unavailable in formula feeding [61]. The WHO strongly recommends breastfeeding children immediately after birth, exclusively for the first 6 months of life, and continuing for at least the first 2 years [62]. Breastfeeding practices in Indonesia vary significantly by region and are influenced by various contextual factors. This heterogeneity underscores the importance of tailored strategies to promote optimal breastfeeding behaviors [63,64]. Strengthening early initiation and sustained breastfeeding should remain a central focus of maternal and child health programs to improve the survival of children with LBW.

Children born with LBW whose mothers had fewer than four ANC visits were found to have an increased risk of mortality compared to those whose mothers had four or more visits. LBW children are biologically more vulnerable. When combined with limited antenatal care (ANC) visits, which reduce opportunities for early risk detection and timely interventions, their risk of mortality increases significantly [65]. Research has shown that attending ANC at least four times can significantly reduce the likelihood of LBW in newborns and enhance child survival rates [66,67,68,69]. ANC provides a vital opportunity for risk screening, counselling, psychosocial support, and targeted interventions aimed at reducing adverse maternal outcomes and improving neonatal health and preparedness, especially for pregnancies at risk of LBW [70,71]. Following the WHO’s recommendations, the Indonesian Ministry of Health has updated its guidelines for integrated ANC by increasing the minimum number of visits from four to six during pregnancy. These visits are distributed across all trimesters and include at least two consultations with a physician [35,72]. However, despite this improvement in policy, the proportion of pregnant women completing the recommended ANC schedule remains suboptimal, indicating persistent gaps in service utilization and access [7]. To address this, context-specific strategies that enhance accessibility, education, and community-based engagement are essential to improve ANC attendance and maximize its protective effect on LBW-related outcomes by strengthening primary healthcare services and ensuring continuity of care.

Unexpectedly, LBW children born in health facilities had a higher risk of mortality than those born at home or in other settings. This surprising finding may be explained by referral bias, where high-risk pregnancies are more likely to be referred to health facilities, as well as disparities in the quality of care [73]. Evidence indicates that only well-equipped medical centers consistently reduce neonatal mortality, while lower-level facilities such as clinics or district hospitals may lack the capacity to manage high-risk births effectively [74,75]. Therefore, delivering in a health facility does not guarantee ensure better outcomes for children born with LBW. Instead, the quality and readiness of the facility are critical determinants of survival. In Indonesia, delivery services are structured into levels: Level I (e.g., Puskesmas and midwife clinics) provide basic care for normal deliveries, while Levels II and III facilities (hospitals) provide specialized care essential for high-risk cases. This distinction is crucial, as the survival of LBW infants often depends on the level of care available at the place of delivery [31,76,77]. Thus, improving LBW survival requires not only promoting institutional deliveries but also strengthening the quality and readiness of all levels of delivery care.

Children with LBW living in rural areas initially demonstrated better survival outcomes compared to those in urban areas, indicating a potentially lower mortality risk. This finding may be due to stronger community support systems [78], higher breastfeeding rates [79,80], or possible underreporting of deaths in rural areas [81]. However, when other variables were considered, the survival difference between rural and urban children was no longer statistically significant. This suggests that socioeconomic factors, access to healthcare, and environmental influences likely responsible for the initial disparity in survival rates, rather than place of residence alone.

Several variables that were statistically significant in the bivariate analysis—such as maternal age at delivery, delivery complications, place of delivery, and residence—lost their significance in the multivariable Cox regression. This reduction in significance may be due to shared variance or confounding effects with other covariates in the model. When these variables are adjusted for simultaneously, their independent effects may diminish, highlighting the complex interplay of factors influencing under-five mortality.

Other variables, such as child’s gender, maternal education and empowerment, and household wealth, were not significantly associated with under-five mortality. This suggests that in the context of children born with LBW survival, maternal biological conditions and health service factors may have a greater influence than sociodemographic characteristics. Although disparities in survival were observed across child, maternal, health service, and household factors, the overall prognosis for children born with LBW was relatively favorable. The inability to estimate median survival time—because survival probabilities did not fall below 50% in most groups—indicates that most children born with LBW survived beyond the follow-up period. Nonetheless, further studies incorporating quality of care and behavioral dimensions are needed to understand these patterns better.

### 4.2. Implications for Practice and Policy

These findings have important implications for nursing and public health policy in Indonesia. Health professionals should prioritize risk screening, promote early and sustained breastfeeding, and ensure high-quality perinatal care to enhance outcomes. At the policy level, efforts must go beyond promoting institutional births to strengthening the readiness of all facility levels, improving ANC access, and ensuring skilled delivery management.

### 4.3. Strengths and Limitations

This study provides valuable insights into the survival of under-five children with LBW in Indonesia by utilizing the 2017 IDHS. This dataset is nationally representative and methodologically rigorous. The study employs time-to-event analysis through Kaplan–Meier estimation and Cox regression, offering a strong statistical approach rarely used in LBW research within Indonesia. Additionally, by integrating child, maternal, household, and health service variables, the study allows for a comprehensive understanding of the factors influencing survival. Identifying modifiable predictors, such as exclusive breastfeeding, antenatal care, and institutional delivery, presents actionable targets for nursing and public health interventions.

However, this study is not without limitations. Although the 2017 IDHS remains the most comprehensive and publicly available dataset, it is not the most recent and may not fully capture current trends. The reliance on secondary data limits the availability of specific clinical or behavioral risk factors, reduces control over the original measurement quality, and the retrospective nature of data collection introduces potential recall bias. The cross-sectional and observational design restricts the ability to draw causal inferences from the associations found. Several predictors showed wide confidence intervals, indicating limited statistical power or potential multicollinearity among the independent variables, though formal diagnostics for collinearity were not performed. Additionally, referral bias might influence the relationship between place of delivery and child mortality, as health facilities often handle higher-risk pregnancies. Further research utilizing more updated, longitudinal, or mixed-methods data is necessary to gain deeper insights into the underlying mechanisms and time-related dynamics.

### 4.4. Future Research

Further research is needed to explore the quality of care, contextual practices, and long-term outcomes, including nutrition and growth while accounting for socioeconomic factors to address disparities and inform targeted interventions.

## 5. Conclusions

This study highlights that maternal age, ANC, delivery complications, place of delivery, and especially breastfeeding—as the strongest predictor—significantly influence the survival of under-five children with LBW in Indonesia. Despite disparities, most children with LBW survived beyond the follow-up period, indicating a generally favorable prognosis.

## Figures and Tables

**Table 1 nursrep-15-00238-t001:** Under-five survival distribution by selected factors and log-rank test results (*n* = 625).

Variable	Category	Deaths (*n*, %)	Survivals (*n*, %)	Total (*n*)	Survival Rate (%)	Mean Survival Time (Month)	χ^2^	*p*-Value
Child factor								
Gender	Male	24 (8.0%)	275 (92.0%)	299	92.0	54.72	1.448	0.229
	Female	17 (5.2%)	309 (94.8%)	326	94.8	55.98		
Maternal factors								
Age at delivery	<35 years	23 (5.5%)	399 (94.5%)	422	94.5	55.99	2.830	0.093 ^†^
	≥35 years	28 (8.9%)	185 (91.1%)	203	91.1	54.17		
Educational level	Elementary	14 (5.5%)	240 (94.5%)	254	94.5	56.02	1.148	0.563
	Secondary/high	24 (7.7%)	289 (92.3%)	313	92.3	54.89		
	Dipl./university	3 (5.2%)	55 (94.8%)	58	94.8	51.79		
Empowerment	Low	6 (9.5%)	57 (90.5%)	63	90.5	54.23	3.416	0.181
	Medium	6 (3.8%)	153 (96.2%)	159	96.2	56.96		
	High	29 (7.2%)	374 (92.8%)	403	92.8	54.85		
Delivery complications	Yes	32 (8.0%)	369 (92.0%)	401	92.0	54.52	4.075	0.044 *
	No	9 (4.0%)	215 (96.0%)	224	96.0	56.95		
Breastfeeding history	Yes	14 (2.5%)	541 (97.5%)	555	97.5	57.60	118.443	0.000 *
	No	17 (38.6%)	43 (61.4%)	70	61.4	41.90		
Health service factors								
ANC history	<4 times	8 (10.1%)	71 (89.9%)	79	89.9	53.39	2.778	0.096 ^†^
	≥4 times	33 (6.0%)	513 (94.0%)	546	94.0	55.68		
Place of delivery	Health facilities	34 (7.2%)	437 (92.8%)	471	92.8	54.86	2.822	0.093 ^†^
	Elsewhere	7 (4.5%)	147 (95.5%)	154	95.5	56.88		
Household factors								
Wealth index	Poor	15 (4.4%)	325 (95.6%)	340	95.6	56.42	4.216	0.121
	Middle	9 (8.8%)	93 (91.2%)	102	91.2	54.45		
	Rich	17 (9.3%)	166 (90.7%)	183	90.7	54.23		
Residence	Urban	26 (8.5%)	279 (91.5%)	305	91.5	54.36	3.860	0.049 *
	Rural	15 (4.7%)	305 (95.3%)	320	95.3	56.38		

^†^ *p* < 0.10, ^*^
*p* < 0.05.

**Table 2 nursrep-15-00238-t002:** Cox proportional hazards regression for under-five mortality among children with low birth weight (*n* = 625).

Variable	HR (Exp (B))	95% CI for HR	*p*-Value
Child factor			
Gender (male)	1.61	0.83–3.11	0.158
Maternal factors			
Age at delivery (≥35)	0.70	0.37–1.34	0.283
Education (ref: elementary)			
Secondary/High	1.16	0.56–2.41	0.698
Dipl./University	1.21	0.31–4.64	0.784
Empowerment (ref: low)			
Medium	0.70	0.21–2.34	0.560
High	0.99	0.38–2.56	0.978
Delivery complications (yes)	1.67	0.75–3.75	0.211
Breastfeeding history (no)	14.58	7.22–29.45	<0.001 *
Health service factors			
ANC history (<4 times)	2.40	0.94–6.11	0.067 ^†^
Place of delivery (health facilities)	2.39	0.88–6.48	0.088 ^†^
Household factors			
Wealth index (ref: rich)			
Poor	1.03	0.42–2.53	0.955
Middle	1.75	0.75–4.09	0.197
Residence (rural)	1.25	0.58–2.67	0.573

HR = hazard ratio; CI = confidence interval. ^†^
*p* < 0.10 (exploratory threshold) and * *p* < 0.001 (statistically significant).

## Data Availability

The data used in this study are available from the DHS Program (www.dhsprogram.com) upon request.

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
