# Peer review of "Predictors of Survival in Under-Five Children with Low Birth Weight: A Population-Based Study in Indonesia"

_nursrep, 2025, doi:10.3390/nursrep15070238_

Round 1

Reviewer 1 Report

Comments and Suggestions for Authors

Dear Authors,

Thank you for your submission. The manuscript covers an important topic; however, I would like to offer the following suggestions to improve its clarity and scientific rigor:

  1. Abstract: Please avoid using abbreviations. Each abbreviation should be written out fully with its definition the first time it appears.

  2. Introduction: Several statements need proper references:

    • The slowdown in U5MR (under-five mortality rate) reduction globally.

    • The neonatal mortality rate in Southeast Asia.

    • The impact of breastfeeding and immunization on child survival.

  3. Discussion: You mention that children with LBW and fewer than four ANC visits had higher mortality risk. Please elaborate on the potential reasons for this association to strengthen the interpretation.

  4. Limitations: Consider adding a dedicated section on study limitations to acknowledge potential constraints and guide future research.

  5. References:

    • Reference 28 and 32 need to be revised.

    • Please ensure consistency in formatting across all references, according to the journal’s reference style.

Best regards,

Reviewer 2 Report

Comments and Suggestions for Authors

This manuscript, titled "Predictors of Survival in Under-Five Children with Low Birth 2 Weight: A Population-based Study in Indonesia", is interesting and has significant potential to contribute to the scientific community. The manuscript is well-organized and thoughtfully written. After a comprehensive and critical evaluation, I wish to bring attention to several findings that can enhance the clarity and impact of their research work. And, I highly recommend publishing this article once the corrections are implemented.

Comments:

  1. Line 19- include the age in a number in parentheses after “under-five mortality”.
  2. The introduction might benefit from clearly distinguishing between neonatal vs. under-five mortality trends.
  3. The cross-sectional design is not ideal for survival analysis, which inherently assumes time-to-event data. Although IDHS includes retrospective data, this should be acknowledged and justified more clearly.
  4. Handling of missing data is not described in detail—what method was used for the 82 excluded?
  5. State whether the proportional hazards assumption was tested (beyond saying it was)—was a Schoenfeld residual test used?
  6. In the statistical analysis section, Model selection is not explained—was it backwards, forward, or based on a theoretical framework?
  7. Consider stating how multicollinearity was checked among independent variables.
  8. The manuscript would benefit from reporting measures of model fit or predictive power (e.g., Harrell’s C-index or AIC).
  9. Some variables significant in bivariate (e.g., residence) lost significance in Cox regression. This deserves explicit comment and interpretation in the results or discussion.
  10. The finding that facility births are associated with higher mortality is likely confounded by referral bias—it’s mentioned in the discussion, but should also be acknowledged as a potential limitation in the results interpretation.
  11. Confidence intervals for some predictors (e.g., place of delivery: 0.88–6.48) are very wide, suggesting low power. Consider stating variance inflation or collinearity diagnostics.
  12. The counterintuitive findings (e.g., facility births) are explained post hoc. Consider adding a stratified analysis or interaction term (e.g., facility quality × birth outcome).
  13. The discussion does not quantify the effect sizes enough (e.g., NNT or attributable risk for breastfeeding).
  14. Consider discussing possible recall bias in maternal reporting of breastfeeding and complications.
  15. The result and discussion sections are well written.
  16. Please include clear and specific policy recommendations based on the results.

Reviewer 3 Report

Comments and Suggestions for Authors

Recommendations and suggestions
In order to have a high degree of novelty, I would propose to the authors an analysis of the nutritional status of children by age groups, not just breastfeeding, as follows: assessment of nutrition from 6 months to 12 months, from 12 months to 24 months and from 24 months to 36 months, reported to the nutritional recommendations for age with the identification of nutritional factors involved in proper growth and development. Child nutrition from 1 year to 3 years is of great importance in the analysis of the rate of undernutrition and infant mortality, representing possible variables of the study lines 125-127 (especially from Section 6 - child health and nutrition), especially since it talks about survival time between 0-4 years (line 126). The results could be correlated with sociodemographic and household characteristics according to the wealth index (as specified in row 140), providing high scientific solidity and much greater interest for readers and could also provide data on intervention measures to reduce mortality risk in disadvantaged regions.

Reviewer 4 Report

Comments and Suggestions for Authors

The authors do not clearly explain how the children were identified within the database.
They should specify which variables from the IDHS 2017 were used to define the children's age and birth-related data.
As data from the IDHS 2017 generally require the use of sampling weights, this information should be explicitly stated.
The Discussion section should acknowledge that this study is a secondary analysis of previously collected DHS data, which comes with inherent limitations (e.g., lack of control over measurement quality, absence of certain clinical variables).
The authors do not provide concrete implementation recommendations (e.g., what measures could improve access to antenatal care or maternal education). Clear and feasible public health action points based on the study’s findings should be included at the end of the Discussion section.
The Conclusion should be structured into: key findings, practical/policy implications, and potential directions for future research. The limitations of the study should also be explicitly stated.

Reviewer 5 Report

Comments and Suggestions for Authors

Under five mortality is a major issue in most developing countries, though in different countries the factors modifying the same may vary. The authors have studied factors affecting mortality in low-birth-weight children in Indonesia. There are however some concerns. The data analysed are from 2012 to 2017, a good 8 years old,  and under 5  mortality has dropped from 24.5/1000 in 2018 to 20.6 /1000 in 2023, as per the World Bank data of June 2025. Today the U5MR is probably below 20/1000 live births. Studying a rapidly changing statistic of 7 years ago makes little sense. The authors themselves acknowledge the limitations of the work stating “Although the 2017 IDHS remains the most comprehensive and publicly available dataset, it is not the most recent and may not fully capture current trends.”

The study has confirmed what has been known since long that maternal age, ANC, delivery complications, place of delivery, and especially breastfeeding—are the strongest predictors of  U5MR. These factors are conventionally known as the key factors in infant mortality the world over. It is doubtful if they need validation. At the planning stage, investigators should ask themselves a few questions. The most important being, are we likely to uncover something new or are we trying to confirm existing knowledge. Some factors may vary across geographies, while others are universal. Studies across different geographies are warranted if there is adequate reasons to believe that factors may differ in the particular geography. Nonetheless, one hopes this paper leads to a further reduction of infant mortality in Indonesia and elsewhere.
